# *Mobiluncus curtisii* Bacteremia: Case Study and Literature Review

**Cade Arries \* and Patricia Ferrieri**

Department of Laboratory Medicine and Pathology, University of Minnesota Medical School, Minneapolis, MN 55455, USA; ferri002@umn.edu
\* Correspondence: arrie003@umn.edu

**Abstract:** Background: There are few reports of bacteremia caused by *Mobiluncus curtisii* in the literature. We present a review of the literature in addition to a case study. Method: We describe the case of an 82-year-old patient who underwent gastrointestinal surgery and subsequently presented with dehydration, nausea, and hyperkalemia secondary to diarrhea. Further clinical work included blood cultures, and the patient was started empirically on piperacillin/tazobactam. Results: After five days, the blood culture bottle showed growth of a gram-variable, curved rod-shaped organism. After culture under anaerobic conditions on sheep blood agar, the organism was identified as *Mobiluncus curtisii* by MALDI-TOF mass spectrometry and enzymatic technology. A review of the literature reveals five additional cases of *Mobiluncus curtisii* bacteremia. Conclusions: This is the sixth case in the literature describing *Mobiluncus* species bacteremia. This organism is rarely identified in blood culture and is most often thought of in the context of bacterial vaginosis. However, the reported cases of bacteremia show gastrointestinal symptoms and presumed gastrointestinal source of infection. The pathogenesis of infection of this organism requires further investigation.

**Keywords:** *Mobiluncus curtisii*; bacteremia; microbiology; anaerobic; blood culture

## 1. Introduction

*Mobiluncus* species are gram variable curved rod-shaped organisms. They are non-spore-forming, motile, with somewhat tapered ends, and may present as pairs with a "gullwing" appearance when two organisms are seen in a pair with tapered ends resembling a seagull shape. In the literature, *Mobiluncus* species are most commonly discussed in relation to their identification in gynecologic infections, in particular, bacterial vaginosis. However, the role of this organism in bacterial vaginosis remains poorly understood. A puzzling finding is that even with the knowledge that this organism is well documented to be resistant to metronidazole, metronidazole remains a common and effective treatment for bacterial vaginosis. For an organism that is resistant to metronidazole, the treatment of bacterial vaginosis with metronidazole and its subsequent resolution would suggest that *Mobiluncus* is not the only organism involved in bacterial vaginosis, and there are likely many complex variables that lead to *Mobiluncus* species identification in association with genitourinary infections. Further research has suggested that although most commonly associated with genital infections, it may be a colonizer of the gastrointestinal tract. This is supported by cases described in a review of the literature presented here and based upon studies that tested rectal samples of patients with and without bacterial vaginosis. It is rare to isolate *Mobiluncus* species from an extragenital site, and here we present a case of *Mobiluncus* bacteremia in an 82-year-old female and a review of the literature regarding *Mobiluncus* bacteremia. Our patient is the sixth reported case in the literature of *Mobiluncus* bacteremia, helping to expand upon the limited literature surrounding this organism in the context of isolating it in blood cultures. Increased awareness of this organism in blood cultures is required to elicit more cases and to form meaningful conclusions about the virulence, origin, and clinical presentation for patients presenting with *Mobiluncus* bacteremia.

## 2. Clinical Presentation

Our patient is an 82-year-old female with a history of diabetes mellitus type 2, hypertension, coronary artery disease, and chronic kidney disease. She had a prolonged hospitalization three months prior to the reported bacteremia due to a small bowel bleed caused by angiodysplasia. She underwent a small bowel resection procedure with ileostomy formation followed by multiple recurrent gastrointestinal operations due to post-operative complications. More recently, she presented to the Emergency Department with complaints of diarrhea and dehydration three months after her hospitalization for the small bowel resection. After initial emergency room testing, she was transferred to a nearby hospital due to an initial finding of hyperkalemia. Of note, two initial blood cultures were drawn in the Emergency Department. After transfer to the hospital, along with the hyperkalemia, the patient was noted to have laboratory findings consistent with acute tubular necrosis/acute renal failure presumably due to severe volume depletion as a result of ileostomy/diarrheal loss. Rehydration with intravenous fluids normalized her renal function. She was discharged nine days after an emergency department visit and hospital admission. Importantly, the prior bowel resection procedure left the patient with a large midline, recovering incision, which was noted to be healing with a wound vac in place at the time of this hospital admission. Urinalysis at the hospital showed many bacteria and a large amount of blood. Computed tomography (CT) imaging showed findings consistent with cystitis, and the patient was empirically started on piperacillin/tazobactam. The presumption was that the patient was septic from a urinary tract infection.

## 3. Materials and Methods

The blood cultures collected at the Emergency Room on initial presentation were documented as a gram-positive rod on Gram's stain. The Gram's stain result was faxed and called to the hospital five days into the admission. The blood culture bottle was then received a day after the Gram's stain result was called at the University of Minnesota Infectious Diseases Diagnostic Lab as an anaerobe referral for definitive identification. The organism was cultured on a blood agar plate under anaerobic conditions (Figure 1a), and the colony Gram's stain showed gram-variable, curved rod-shaped organisms (Figure 1b). The organism was identified as *Mobiluncus curtisii* by MALDI-TOF mass spectrometry (Vitek MS, bioMérieux, Marcy l'Etoile, France) at a confidence level of 99.9% and as *Mobiluncus curtisii* by enzyme technology using the RapID™ ANAII system for anaerobic bacteria (ThermoFisher Scientific, Waltham, MA, USA) at a probability of >99.9%. A limitation of our study is that data from sequencing are not available, as it is not routinely performed for identification in our laboratory. Therefore, it is highly probable, but not 100% certain, that this organism was correctly identified as *Mobiluncus curtisii* by two separate testing modalities.

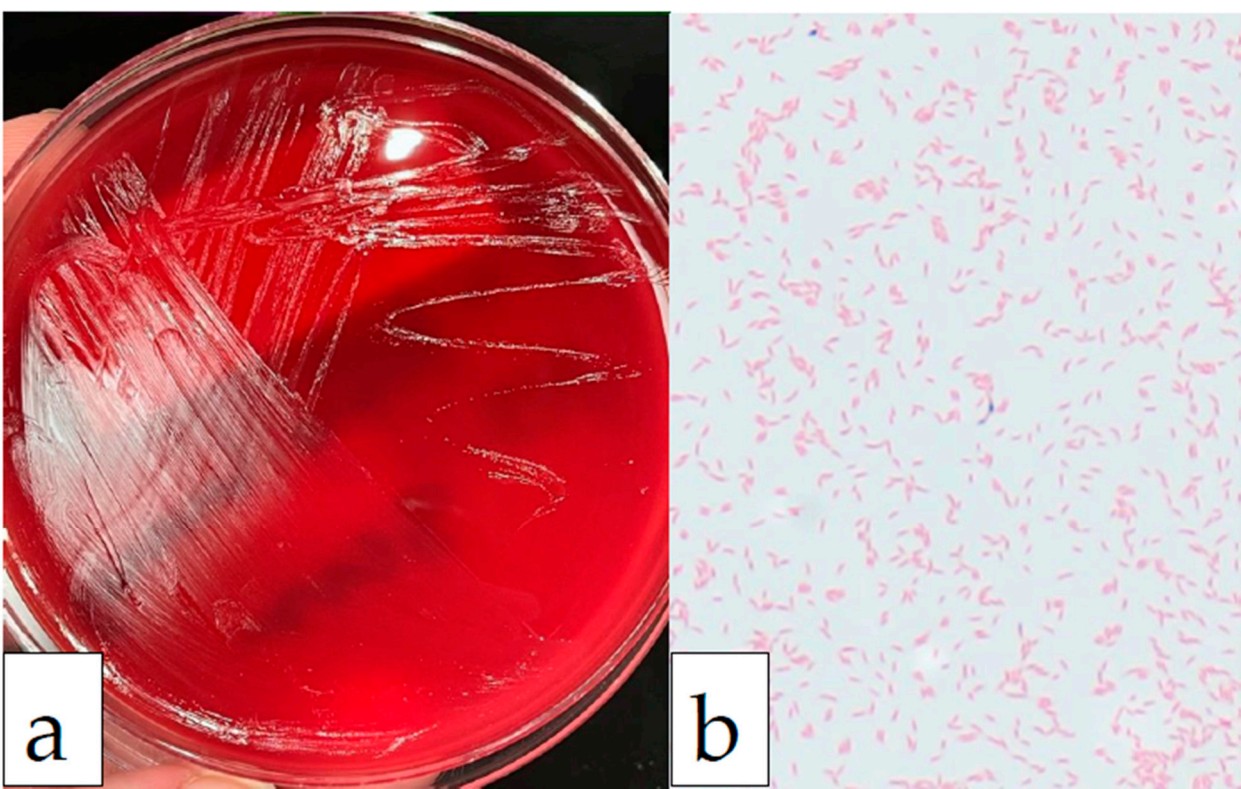

**Figure 1.** (**a**) Image of the sheep blood agar plate grown in anaerobic conditions showing colorless, translucent, smooth, convex colonies after 5 days of incubation. (**b**) Image of Gram's stain at 1000× magnification, with oil immersion. The organism was curved and rod-shaped, gram-variable, and a predominant gram-negative staining pattern was displayed here.

## 4. Discussion

The *Mobiluncus* genus is subdivided into two species, *Mobiluncus mulieris* and *Mobiluncus curtisii* (which is even further subdivided into subspecies *Mobiluncus curtisii* subsp. *curtisii* and *Mobiluncus curtisii* subsp. *holmesii*) [1]. *Mobiluncus curtisii* subspecies *curtisii* is typically a gram-variable, curved rod-shaped, non-spore-forming, motile organism with tapered ends, occurring singly or in pairs with a "gullwing" appearance. Clusters of two to six flagella which are longer than the bacterial cells, have been observed on electron micrographs. Although the Gram's stain reactions of the curved, rod-shaped organisms are reported as variable, electron micrographs reveal multilayered gram-positive cell walls lacking an outer membrane. Even in this case, the initial callback stated that the organism was gram-positive, whereas after identification at the University of Minnesota, the organism was identified as gram-variable, but as seen in Figure 1b, shows a predominantly gram-negative type staining with only rare gram-positive organisms. *Gardnerella vaginalis*, another gram-variable organism, also has a multilayered wall. The thinness of the peptidoglycan layer may explain the tendency of the curved rod-shaped organisms to stain gram-negative. Based upon observations of this organism in culture, young cultures tended to show more gram-positive staining [2]. The apparent absence of an outer membrane suggests that *Mobiluncus* species more closely resemble gram-positive organisms than gram-negative organisms. This conclusion is supported by reports that these organisms are resistant to colistin and nalidixic acid and are susceptible to penicillin, ampicillin and vancomycin. Hydroxy fatty acids, commonly found in gram-negative cell walls, are absent in these organisms [2]. Moreover, DNA sequencing places the organism in the family *Actinomycetaceae*, which is a family of gram-positive organisms [3].

*Mobiluncus* species have been shown to be associated with isolates from patients with bacterial vaginosis [4]. Because *Mobiluncus curtisii* and *Mobiluncus mulieris* do not produce

putrescine or cadaverine in peptone-starch-dextrose broth supplemented with lysine and ornithine, these species are not the source of the amine odor associated with bacterial vaginosis [2]. One published study by author Holst [5] showed that *Mobiluncus* species were isolated from 97% of patients with bacterial vaginosis. A later study by Schwebke et al. [6] showed a similar result using polymerase chain reaction (PCR) that *Mobiluncus* species were identified in 84.5% of women with bacterial vaginosis and only 38% of women without bacterial vaginosis infection. Moreover, *Mobiluncus curtisii,* specifically, was rarely detected in women without bacterial vaginosis, and it was found in 65.3% of women with bacterial vaginosis [7]. The pathogenic role of *Mobiluncus* is still unclear in bacterial vaginosis since *Mobiluncus* species are usually resistant to metronidazole, which is an effective treatment for most cases of bacterial vaginosis [6]. Holst [5] also showed, of those women with bacterial vaginosis, 45 to 62% had *Mobiluncus* species isolated from concurrent rectal samples, whereas *Mobiluncus* was isolated from only 10 to 14% of the rectums of women without bacterial vaginosis.

Additionally, Holst [5] reported that 5–11% of men and children had *Mobiluncus* species isolated from rectal samples. These findings caused researchers to deliberate if this organism resides in the gastrointestinal tract and may not be primarily thought of as a genital bacterium. If the intestinal tract is the main reservoir for *Mobiluncus* species, it could be presumed that an acute abdominal problem may allow dissemination and initiate bacteremia. Alternatively, as a possibility, in this case, the bacteria may have spread from the genitourinary space or the ileostomy site to nearby damaged skin or the relatively nearby large midline surgical incision. Given the extensive gastrointestinal surgeries and healing ileostomy site, the presumed source, in this case, is gastrointestinal; however, travel from the genitourinary space to nearby damaged epithelium cannot be entirely ruled out.

Extragenital infection by *Mobiluncus* is rare and has been reported in breast abscesses and bacteremia [1]. To our knowledge, this is only the sixth case of *Mobiluncus* bacteremia reported in the literature (Table 1). Four patients, including ours, had *Mobiluncus curtisii* identified. Of the five cases of *Mobiluncus* bacteremia reported in the literature, four cases were female. The average age of the reported patients, including our case, is 51 years, with a range from 35 years to 82 years. Two cases of *Mobiluncus* bacteremia showed life-threatening (severe hypotension) or fatal infections. One patient diagnosed with *Mobiluncus* bacteremia did not survive, related to a massive intracerebral hemorrhage of unknown cause. One patient had no reported comorbidities suggesting that this unusual bacteremia can affect people who are presumably immunocompetent. Most of the reported cases presented with fever, and all cases had some report of gastrointestinal disturbances, including nausea, diarrhea, vomiting, or abdominal pain. The presumed origin, based upon the case reports of the bacteremia, was the gastrointestinal tract in four patients, including this patient [8–10], with the other two sources presumed to be uterine [11] and unknown [1]. Treatment with an antibiotic led to full recovery in all but one patient. In our case, the patient was treated empirically with piperacillin/tazobactam for a presumed urinary tract infection. The only organism identified by microbiology culture was on the blood culture from the Emergency department, which was identified as *Mobiluncus curtisii* by the Infectious Diseases Diagnostic Lab at the University of Minnesota. In the reported cases, the blood culture bottle signaled a positive culture in as little as 24 h to as much as 9 days; this may be related to bacterial load in each individual. The person with *Mobiluncus* bacteremia who did not recover had a positive blood culture after 5 days, which would not support a correlation between the time to a positive culture and clinical course. However, the patient who presented with severe hypotension had a positive blood culture signal after 24 h of incubation, so there may be a relationship between clinical presentation and time to positive culture for this organism. Our understanding of this relationship will be improved as more case reports are published. For those cases with reports of antibiotic resistance, most showed resistance to metronidazole, consistent with the literature and are susceptible to many different antibiotics. The case of *Mobiluncus* bacteremia caused by *Mobiluncus mulieris* was sensitive to metronidazole, consistent with what is reported in the

literature. Interestingly, three of the five reported cases had metabolic panels which showed transaminitis, and one patient even presented with jaundice and hepatomegaly. This could indicate that *Mobiluncus* species may cause liver injury. As more cases continue to be reported in the literature, our ability to draw meaningful conclusions regarding infection with this organism will improve tremendously.

The pathogenesis of infection with and virulence factors of *Mobiluncus curtisii* are still not completely understood. One study, by author Zeng [12], showed that in the *Mobiluncus curtisii* genome, there is a genomic fragment encoding a 25 kDa pore-forming toxin, the CAMP factor, which is known to be involved in the synergistic lysis of erythrocytes, namely the CAMP reaction. Additionally, another study by Menolascina et al. [13] suggested that both *Mobiluncus curtisii* and *Mobiluncus mulieris* cells have the capacity to adhere, mediated by adhesin, in the absence of glucose or mannose. Lastly, Taylor-Robinson et al. [14] showed that the centrifuged supernatants of cultures of *Mobiluncus curtisii* and *Mobiluncus mulieris* had a toxic effect on epithelial cells, corroborated by the Menolascina et al. group. These factors taken together, in the appropriate growth conditions, could allow *Mobiluncus curtisii* to adhere to and destroy epithelial cells allowing entry of the organism into the bloodstream from a source of colonization, either the gastrointestinal tract or the genitourinary tract. As more cases of *Mobiluncus* species bacteremia are published we will begin to understand more of the ecology and the pathogenesis of infection with this organism.

**Table 1.** Clinical and laboratory features of reported Mobiluncus bacteremia cases.

| Age, Sex [Reference] | Organism Identification | Incubation Time to Positive Blood Culture Notification | Presumed Bacteremia Origin | Susceptibilities | Presenting Signs and Symptoms | Comorbid Conditions | Lab Tests | Outcome |
|---|---|---|---|---|---|---|---|---|
| 39, F [8] | *Mobiluncus* spp. | 9 days | G.I. tract | S: Penicillin Gentamicin, Clindamycin; R: Metronidazole, cotrimoxazole; NK: Cefazolin | Confusion Abdominal pain Diarrhea Jaundice Hepatomegaly Fever (38 °C) | Alcoholic Liver disease | WBC: 5.8 Plt 90 Hgb: 9.1 Metabolic panel: Transaminitis | Recovery |
| 33, F [11] | *Mobiluncus curtisii* | 3 days | Uterus | S: Amoxycillin, Cefoxitin, Gentamicin, Clindamycin; R: Metronidazole | Headache Abdominal pain Vomiting. Fever (38.8 °C) | Spontaneous abortion Hepatitis B Streptococcal antigen positive | WBC: 9.8 Plt: 153 Hgb: 10.9 Metabolic Panel: within normal limits | Recovery |
| 62, F [9] | *Mobiluncus mulieris* | 2 days | GI tract (Perforated colon, intraabdominal abscesses) | S: Metronidazole; R: Not stated | Confusion Abdominal pain Vomiting Afebrile tachycardia (120 bpm) | Diabetes, atrophic uterus | WBC: 21.1 Metabolic Panel: Not stated | Recovery |
| 54, F [1] | *Mobiluncus curtisii curtisii* | 24 h | Not known | S: Penicillin, Cefotaxime; R: Metronidazole | Confusion Headache Abdominal pain Vomiting Fever (38.3 °C) Hypotension (56/35 mmHg) Tachycardia (120 bpm) | None | WBC: 4.8 Plt: 39 Hgb: 12.9 Metabolic Panel: Transaminitis | Recovery |
| 35, M [10] | *Mobiluncus curtisii* | 5 days | GI tract (ulcerative colitis) | Not stated | Anorexia Weakness Falls Unconscious (massive left sided intracerebral hemorrhage) Fever (>38.5 °C) | Ulcerative colitis alcoholism | WBC: 7.5. Plt: 44. Hgb: not stated Metabolic Panel: Transaminitis | Death |
| 82, F (our case) | *Mobiluncus curtisii* | 5 days | GI tract (angiodysplasia status post small bowel resection and ileostomy) | S: Penicillin, Clindamycin, Cefotaxime, Meropenem, Amoxicillin/Clav. R: Metronidazole | Nausea Dehydration Diarrhea Hyperkalemia Cystitis Afebrile | Diabetes angiodysplasia of bowel (status post resection, ileostomy) | WBC: 16. Plt: 437 Hgb: 13.2 Metabolic Panel: Renal Failure | Recovery |

WBC, peripheral blood white blood cell count ($\times$ 109/L); Plt, platelet count ($\times$ 109/L); Hgb, hemoglobin concentration (g/dL). S, sensitive; R, resistant; NK, not known.

## 5. Conclusions

Our case is only the sixth case reported in the literature describing *Mobiluncus* species bacteremia. *Mobiluncus curtisii* is rarely identified in blood culture and is most often thought of in the context of bacterial vaginosis. However, the patients with *Mobiluncus* bacteremia most often present with gastrointestinal symptoms including diarrhea, abdominal pain, and transaminitis, and the gastrointestinal tract is presumed to be the source of infection. The pathogenesis of infection with this organism requires further investigation. Continued attention to this organism will inevitably lead to new case reports and further understanding of this rarely reported bacteremia with *Mobiluncus* bacterial species.

**Author Contributions:** Conceptualization, C.A. and P.F.; writing—original draft preparation, C.A.; writing—review and editing, C.A. and P.F. All authors have read and agreed to the published version of the manuscript.

**Funding:** This research received no external funding.

**Informed Consent Statement:** General informed consent was obtained from all subjects involved in the study.

**Data Availability Statement:** Not applicable.

**Acknowledgments:** Infectious Diseases Diagnostic Laboratory (IDDL) staff at the University of Minnesota Medical Center.

**Conflicts of Interest:** The authors declare no conflict of interest.

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
