# Peer review of "Mobiluncus curtisii Bacteremia: Case Study and Literature Review"

_2036-7449, doi:10.3390/idr14010009_

Round 1

Reviewer 1 Report

Dear Authors,

Your manuscript entitled “Mobiluncus curtisii bacteremia, case study and literature review” is valuable and noteworthy.

Indeed, the data available in the literature most often link Mobiluncus genus with gynecological infections, in particular bacterial vaginosis. However, several available research has suggested that these bacteria may be colonizers of the gastrointestinal tract.

It was a very good idea to compare your own case with similar ones described in the available literature. Especially since this is only the sixth case described. This will help expand our understanding of infections caused by Mobiluncus sp.

In my opinion, the corrections listed below should be introduced to the manuscript:

  1. When you quote at the end of a sentence, first, write the citation number in brackets, then the full stop sign at the end of the sentence:

e.g. line 106: Now it is: absent in these organisms.[2] Should be: absent in these organisms [2].

Similarly in lines 108, 111, 117, 119, 123, 134,… etc. – check this in the whole manuscript

  1. When citing a work in the text, enter the citation number in brackets just after the author's name:

Lines 111-112: The sentence is: “One published study, by author Holst, showed Mobiluncus species were isolated from 97% of patients with bacterial vaginosis. [4]” Should be: One published study, by author Holst [4], showed Mobiluncus species were isolated from 97% of patients with bacterial vaginosis.

Similarly in lines  119, 122,… etc. – check this in the whole manuscript

  1. line 136 – „identified” without Italics
  2. line 171 – delete the word „author”
  3. References:
  • change the way of writing the authors names, name of the journal, year, volume, in accordance with the MDPI guidelines for authors
  • list all the authors instead of writing „et al.”
  • lines 208, 220 – please correct the typos in the titles of the articles
  • use italics when writing the name of the species or genus

6.  Moreover, Authors state that the Table in the manuscript is adapted from Table created by Hill et al. [1]. However, the manner of Table composition and content is identical to the one prepared by Hill et al. Have Authors received permission? If not, the table should be reorganized.

Best regards,

Author Response

REVIEWER 1:

Thank you so much for your thoughtful review of our manuscript. We appreciate your time and attention to detail. Please find below our responses to your edits and suggestions.

Point 1: When you quote at the end of a sentence, first, write the citation number in brackets, then the full stop sign at the end of the sentence:

  • Response 1: I have edited the document to reflect this correct formatting.

Point 2: When citing a work in the text, enter the citation number in brackets just after the author's name:

  • Response 2: I have edited the document to reflect this correct formatting.

Point 3: line 136 – „identified” without Italics

  • Response 3: Corrected

Point 4: line 171 – delete the word „author”

  • Response 4: Edited.

Point 5: References:change the way of writing the authors names, name of the journal, year, volume, in accordance with the MDPI guidelines for authors:

  • Response 5: I have edited the references according to the MDPI guidelines.

Point 6: lines 208, 220 – please correct the typos in the titles of the articles

  • Response 6: Corrected

Point 7: use italics when writing the name of the species or genus

  • Response 7: Corrected

Point 8: Moreover, Authors state that the Table in the manuscript is adapted from Table created by Hill et al. [1]. The table should be reorganized.

  • Response 8: The table has been restructured and reorganized

Reviewer 2 Report

This article is a case report of a patient with a relatively rare cause of bacteraemia- Mobiluncus  curtisii. The paper reports the infection and compares it to the few known cases previously published in the literature. This is a worthwhile addition to the literature and may prompt further reporting of this pathogen that may increase our knowledge of it.

I have only a couple of minor suggestions to improve the manuscript.

There is a reasonably lengthy discussion over the Gram +ve/Gram variable phenotype of the strain, which could maybe be reduced a little. Previous genome sequencing and analysis puts the Mobiluncus genus in high GC gram + family of ActinobacteriaActinomycetacea. In my opinion it would be worth adding into the description of the organism that DNA sequencing places the genus as a Gram + organism, rather than spending so much time on the variable Gram staining (although the Gram stain is a clinically relevant phenotype).

Page 4 “identified” should not be in italics (line 136).

Author Response

Reviewer #2:

Thank you for your time and attention to our manuscript. We appreciate your helpful suggestions.

Point 1: Regarding your comment: “There is a reasonably lengthy discussion over the Gram +ve/Gram variable phenotype of the strain, which could maybe be reduced a little. Previous genome sequencing and analysis puts the Mobiluncus genus in high GC gram + family of ActinobacteriaActinomycetacea. In my opinion it would be worth adding into the description of the organism that DNA sequencing places the genus as a Gram + organism, rather than spending so much time on the variable Gram staining (although the Gram stain is a clinically relevant phenotype)”

  • Response 1: As you say the “Gram stain is a clinically relevant phenotype.” Therefore, we would like to keep our information because we consider this essential and scholarly, but we will certainly enhance the manuscript by adding your suggestion on the DNA sequencing of the organism and its classification into the gram-positive family of

Point 2: Page 4 “identified” should not be in italics (line 136).

  • Response 2: Corrected.

Reviewer 3 Report

This case report is about a case of Mobiluncus curtisii bacteremia.

1. It is well written, but identification methods with MALDI-TOF and RapID ANA system for this bacteria is not sufficient for recent standard of method. Sequencing is necessary for the correct identification.

2. The case reports are not rare for this strain. Most of the case reports are published 10 years ago, so it may not be so important case report if the case has notable characteristics.

3. Missed case reports are present. Cases might be too enough to be reported, so for the publication, sufficient numbers of these isolates might be needed.

- One case is also missed: Jose Miguel Sahuquillo-Arce. Mobiluncus curtisii bacteremia. Anaerobe 2008.

Author Response

Reviewer #3:

Thank you Reviewer #3 for your review. Please see below for responses to your edits and suggestions.

Point 1. It is well written, but identification methods with MALDI-TOF and RapID ANA system for this bacteria is not sufficient for recent standard of method. Sequencing is necessary for the correct identification.

  • Response 1: We appreciate the contribution of 16s rRNA sequencing, but the organism is no longer available to do this

Point 2. The case reports are not rare for this strain. Most of the case reports are published 10 years ago, so it may not be so important case report if the case has notable characteristics.

  • Response 2: We believe the totality of this patient and  the bacteremia and the Discussion are very  instructive for the audience served by Infectious Disease Reports

Point 3. Missed case reports are present. Cases might be too enough to be reported, so for the publication, sufficient numbers of these isolates might be needed.

One case is also missed: Jose Miguel Sahuquillo-Arce. Mobiluncus curtisii bacteremia. Anaerobe 2008.

  • Response 3: This case, published in Anaerobe in 2008, is included in our table and in our discussion and is seen as the last reference in the references list.

This manuscript is a resubmission of an earlier submission. The following is a list of the peer review reports and author responses from that submission.